# Fractionation of Heavy Metals in Multi-Contaminated Soil Treated with Biochar Using the Sequential Extraction Procedure

**DOI:** 10.3390/biom11030448

**Published:** 2021-03-17

**Authors:** Mahrous Awad, Zhongzhen Liu, Milan Skalicky, Eldessoky S. Dessoky, Marian Brestic, Sonia Mbarki, Anshu Rastogi, Ayman EL Sabagh

**Affiliations:** 1Key Laboratory of Plant Nutrition and Fertilizer in South Region, Ministry of Agriculture, Institute of Agricultural Resources and Environment, Guangdong Academy of Agricultural Sciences, Guangzhou 510640, China; mahrousawad@yahoo.com; 2Department of Soils and Water, Faculty of Agriculture, Al-Azhar University, Assiut 71524, Egypt; 3Department of Botany and Plant Physiology, Faculty of Agrobiology, Food, and Natural Resources, Czech University of Life Sciences Prague, Kamycka 129, 165 00 Prague, Czech Republic; skalicky@af.czu.cz (M.S.); marian.brestic@uniag.sk (M.B.); 4Department of Biology, College of Science, Taif University, P.O. Box 11099, Taif 21944, Saudi Arabia; es.dessouky@tu.edu.sa; 5Department of Plant Physiology, Slovak University of Agriculture, Nitra, Tr. A. Hlinku 2, 949 01 Nitra, Slovakia; 6National Institute of Research in Rural Engineering, Water 13 and Forests (INRGREF). BP 10, Ariana 2080, Tunisia; mbarkisonia14@gmail.com; 7Department of Ecology and Environmental Protection, Poznan University of Life Sciences, Piątkowska 94, 60-649 Poznan, Poland; anshusls@gmail.com; 8Department of Agronomy, Faculty of Agriculture, Kafrelsheikh University, Kafrelsheikh 33516, Egypt

**Keywords:** heavy metals, paulownia, bamboo, biochar, distribution, availability

## Abstract

Heavy metals (HMs) toxicity represents a global problem depending on the soil environment’s geochemical forms. Biochar addition safely reduces HMs mobile forms, thus, reducing their toxicity to plants. While several studies have shown that biochar could significantly stabilize HMs in contaminated soils, the study of the relationship of soil properties to potential mechanisms still needs further clarification; hence the importance of assessing a naturally contaminated soil amended, in this case with Paulownia biochar (PB) and Bamboo biochar (BB) to fractionate Pb, Cd, Zn, and Cu using short sequential fractionation plans. The relationship of soil pH and organic matter and its effect on the redistribution of these metals were estimated. The results indicated that the acid-soluble metals decreased while the fraction bound to organic matter increased compared to untreated pots. The increase in the organic matter metal-bound was mostly at the expense of the decrease in the acid extractable and Fe/Mn bound ones. The highest application of PB increased the organically bound fraction of Pb, Cd, Zn, and Cu (62, 61, 34, and 61%, respectively), while the BB increased them (61, 49, 42, and 22%, respectively) over the control. Meanwhile, Fe/Mn oxides bound represents the large portion associated with zinc and copper. Concerning soil organic matter (SOM) and soil pH, as potential tools to reduce the risk of the target metals, a significant positive correlation was observed with acid-soluble extractable metal, while a negative correlation was obtained with organic matter-bound metal. The principal component analysis (PCA) shows that the total variance represents 89.7% for the TCPL-extractable and HMs forms and their relation to pH and SOM, which confirms the positive effect of the pH and SOM under PB and BB treatments on reducing the risk of the studied metals. The mobility and bioavailability of these metals and their geochemical forms widely varied according to pH, soil organic matter, biochar types, and application rates. As an environmentally friendly and economical material, biochar emphasizes its importance as a tool that makes the soil more suitable for safe cultivation in the short term and its long-term sustainability. This study proves that it reduces the mobility of HMs, their environmental risks and contributes to food safety. It also confirms that performing more controlled experiments, such as a pot, is a disciplined and effective way to assess the suitability of different types of biochar as soil modifications to restore HMs contaminated soil via controlling the mobilization of these minerals.

## 1. Introduction

Heavy metals (HMs) are considered severe worldwide problems that threaten the ecosystem and human life [1,2]. According to the Ministry of Environmental Protection and Ministry of Land and Resources, PRC, many HMs pollution sources and their high level in the soil cause hazardous effects on soil quality, fertility, food safety, and human health [3]. Among these sources, effluents discharges and mining activities for more than 50 years in northern Guangdong Province, China, are mainly implicated in the elevated levels with these pollutants. Shaoguan city is a representative area of these mining activities with high levels of heavy metals and low pH values [4]. However, studies on geochemical fractions of heavy metals in this area are limited.

In the past, various techniques were used to reduce HMs contamination by phytoremediation [5,6]. This idea may be useful in fairly polluted soils, but in highly contaminated soils, a gradual decrease in plant growth makes it ineffective. In this case, biochar serves as a faster, more efficient, and environmentally friendly alternative via immobilization of heavy metals in the agricultural soil [7,8,9,10,11,12]. The characteristics of biochar (surface area, pH, ash, elemental composition, and carbon content) make it one of the most suitable materials to handle and remediate different heavy metals [13]. Biochar properties are highly dependent on the temperature, duration of pyrolysis, feedstock from which the biochar is made, and final acidity [2].

Generally, the total heavy metal concentration is inexpressive of its toxicity, harm to plants and the environment. A low but mobile amount may be more harmful than a large immobile amount depending on the chemical form, species, and solid-phase characteristics, which significantly affect its redistribution [12,14,15]. These forms can be controlled by precipitation, ion exchange, water compounds, stabilization, and plant uptake [12]. Several other critical chemical factors also influence the mobility of soil HMs. One of the dominant factors is the high soil pH that immobilizes metals [16]. Additionally, soil organic matter content and its type, and the synergetic effects of soluble ions are limiting factors in stabilization and metal movement in the contaminated soils [13,17]. Therefore, we foster the hypothesis that biochar is characterized by being a tool capable of raising soil pH. The organic matter makes it a revolutionary solution to make these geochemical fractions more controlled, especially in the acidic soils of Shaoguan.

To fractionate these metals, many authors have recognized that sequential extraction is a crucial method that could provide useful information on the bioavailability of metals and their transformation forms [14,18,19,20]. Although several studies confirmed that biochar is an important material for adsorbing HMs from contaminated soil, the mechanism is still unclear [4,21]. The current study aims to (i) evaluate the effects of biochar types and their application rates on the geochemical form of lead (Pb), cadmium (Cd), zinc (Zn), and copper (Cu) in contaminated soil, and (ii) assess their effects on soil organic matter, soil pH and their relationship among the different forms of HMs.

## 2. Materials and Methods

### 2.1. Soil Samples and Biochar

Surface soil samples (0–20 cm) were collected from Shaoquan city, Guangdong province, north China, which is a representative area for mining activities. In addition, this place has received liquid waste discharges for over 50 years, which causes a high level of heavy metals, especially lead and zinc. This soil is classified according to the soil survey staff [22] as Udult. The collected samples were air-dried, crushed, and sieved through a 2 mm sieve, and then kept for soil analysis. Paulownia (PB) and Bamboo (BB) (by-products of Huanyu Energy Technology Co., Ltd., Guangzhou city, China) were pyrolyzed at 700–800 °C in a EURO-Model CK-2-Charcoal Kiln. A stainless-steel mill crushed the produced aforesaid types of biochar to fixed grain size (<0.1 mm). An amount of 1.5 kg of the tested soil was homogeneously mixed with biochar materials at rates of 2, 4, and 6%, and replicated three times plus a control treatment with zero biochar addition (CK). Through experiment duration (two months), the soil treated with biochar was moist to 70% of its field capacity. The relative humidity during this period ranged between 60 and 70%, and the temperature was 25–30 °C. The biochar’s elemental composition was executed using energy dispersive X-ray spectrometry (EDS) elemental mapping (Figure 1). The functional groups involved in SD adsorption were analyzed by Fourier-transform infrared (FTIR) spectroscopy (Nicolet 6700, Thermo Scientific). After 60 days, the soil reaction (pH) and salinity (electrical conductivity-EC) of the suspension 1:2.5 (soil: water ratio) were determined by a pH- and EC-meter, respectively [23]. Soil organic matter (SOM) content was determined through the oxidation of organic matter by dichromate in an acid medium [24]. Some properties of the soil and biochar types were measured according to [25], which are shown in Table 1. In the end, the heavy metals (Pb, Cd, Zn, and Cu) in the soil sample were extracted using USEPA Toxicity Characteristic Leaching Procedure (TCLP), the method 1311 [26] was used to assess the available form. Their geochemical fractions were also determined according to the European Union Bureau of Reference Procedure (EUBCR) as mentioned below.

### 2.2. USEPA Extraction

In polypropylene centrifuge tubes, 20 mL of unbuffered glacial acetic acid (pH 2.88) were added to one gram soil sample. The mixture was shaken using an orbital shaker at 30 rpm for 18 h, centrifuged at 3000 rpm for 30 min. The supernatant was filtered and quantified with atomic absorption spectroscopy (AA 800, Perkin Elmer, Centennial, CO, USA).

### 2.3. Sequential Extraction Procedure

A sequential extraction procedure was applied according to EUBCR to determine heavy metals redistribution (Pb, Cd, Zn, and Cu) as described by [27]. One gram of soil sample was placed into a 100 mL polycarbonate tube then four sequential extractions were carried out as fractions of F1 (acid-soluble and carbonate bound), then 40 mL of acetic acid (0.11 M) were added, shaken for 16 h, centrifuged at 3000 rpm for 20 min, and finally filtered using Whatman-42 filter paper. F2 (bound to Fe-Mn oxides), to the residual from F1, add 40 mL of hydroxylamine hydrochloride (0.5 M) then the extraction procedure was performed as described in F1. F3 (oxidizable bound of organic matter), to the residual from the F2, add 10 mL H_2_O_2_ (8.8 M), evaporated to near dryness at 85 °C, cooled then 50 mL of 1 M NH_4_OAc (pH 2) was added, and the extraction procedure was followed as described in F1. The residual (remaining residue, bound to the mineral matrix) was determined using HNO_3_-HF-HClO_4_ acids on a hot plate without boiling, filtered, and diluted with pure deionized water to 15 mL. Before the next step, the residue from each stage was washed with 20 mL of de-ionized water and shaken again for 30 min and then centrifuged for 20 min at 3000 rpm. The supernatant liquid was also collected into a 100 mL polyethylene volumetric flask and stored in a refrigerator at 4 °C before analysis. The metal concentrations were determined with atomic absorption spectrophotometer (AA 800, Perkin Elmer Co., 14775 E Hinsdale Ave, Centennial, CO, USA).

## 3. Results

### 3.1. Biochars Description

Depending on SEM-EDS analysis, paulownia biochar (PB) had higher oxygen and lower carbon content than bamboo biochar (BB). The relative Si, K, and Ca contents in the PB were higher than those of the BB, while the BB had higher P and Mg content than PB (Figure 1). Additionally, the paulownia biochar had higher alkalinity than that of bamboo biochar (Table 1). The O/C ratios of PB and BB were 0.22 and 0.19, respectively (Figure 1). The results of the FTIR analysis of PB are shown in the FTIR spectra presented in Figure 2. Stretching of hydroxyl (-OH), carbonyl or carboxyl (C=O), aromatic (C-H) and Si-O functional groups for PB than BB indicate that the ability of paulownia makes it more than bamboo to absorb heavy metals in the soil.

### 3.2. Effects of Biochar on Soil Properties

Data in Table 2 indicated that biochar applications significantly increased the organic matter (OM) content, and the soil pH was raised along with increasing biochar levels compared to the control. The highest PB and BB rates increased the OM by 1.28 and 1.17 times, respectively, over the untreated one. The pH values were increased due to the biochar application, where it reaches the maximum value at 4% PB and 6% BB. The EC varied according to the types of biochar and their application rates. The PB application decreased soil salinity (EC), which was distinct from the low application level. On the other hand, the EC values gradually increased with the increasing BB levels.

### 3.3. Heavy Metals Mobility

The illustrated data in Table 3 shows that the biochar additions caused a decrease in HMs concentration compared with CK treatment. A significant reduction of the TCLP-extractable Pb, Cd, Zn, and Cu in the tested soil was observed using biochar. The magnitude reductions varied with the types of feedstock and their application rates. Generally, the PB material was more efficient in reducing the TCLP-extractable metals than that of BB. As compared to the control, application of 6% PB decreased the TCLP-extractable metals by 22, 17, 24, and 45% for the Pb, Cd, Zn, and Cu, respectively. On the contrary, the corresponding reductions were 20.45, 15, 15, and 12%, respectively, with 6% BB.

### 3.4. Heavy Metals Redistribution

#### 3.4.1. Acid-Soluble Fraction

Applying the biochar decreased the acid-soluble fractions of Pb, Cd, Zn and Cu compared to the control treatment (Table 4). The degree of reduction of these metal fractions was depended on the types of biochar, application rates, and metal species. Adding 6% PB or BB remarkably decreased the Cu content followed by Pb over the control treatment, and their reductions reached 37.89 and 31.22%, respectively. Despite Cd total content (2.5%), its acid-soluble fraction is very high since it reached 64 to 73% of its total content compared to other metals. The acid-soluble fraction varied from 12 to 14% for Pb, 14 to 17% for Zn, and 4 to 6% for Cu, and these variations were governed by the nature of biochar and application rates. The mobility of target metals due to the biochar application was in the decreasing order of Cd > Zn > Pb > Cu.

#### 3.4.2. Bound to Fe and Mn Oxides

The biochar treatments realized a significant reduction in cadmium oxide bounds with few exceptions (Table 5). The PB reduced the Cu bound to Fe/Mn oxides, especially with the highest application rate, while the BB showed an insignificant increase. There was an irregular trend of decreasing Pb oxides bound fraction with either type or level of biochar application. The PB at a 4% level showed the highest Pb concentration bound to the Fe/Mn oxides, while BB applied at a level of 6% showed the lowest one (Table 5). The Pb was predominantly adsorbed by the Fe/Mn fractions (45 to 63%), while an insignificant amount of Cd, Zn and Cu was extracted from the Fe/Mn oxide bound fraction. In Zn case, all the biochar treatments significantly (*p* < 0.05) increased its oxide bound fraction.

#### 3.4.3. Bound to Organic Matter

Biochar application tends to significantly increase in metal concentrations associated with soil organic matter (Table 6). This increase was mainly on the acid extractable (pore water ones) expense and partially on the oxides bound. The organic bound of Pb fraction, as relative change, increased over control by 62 and 61% at the highest PB and BB rates, respectively. Application of PB at 6% increased the OM bound fraction of Cd, Zn, and Cu to reach 62, 34, and 61%, respectively. BB highest addition also increased the organically bound fraction by 49, 42 and 22 % for Cd, Zn and Cu, respectively. The results indicated that Pb was adsorbed mainly by the organic fractions, while the large portion of the acid extractable bound to Cd. Meanwhile, Fe/Mn oxides bound represents the large portion associated with Zn and Cu.

#### 3.4.4. Residual Fraction

The residual fraction of Pb, Cd, Zn, and Cu did not change significantly due to the application of any of the biochar tested with few exceptions for Pb (Table 7). The residual fraction of Pb, Cd, Zn, and Cu varied from 18 to 20%, 5 to 6%, 65 to 67% and 42 to 47%.

#### 3.4.5. Correlation Analysis

As can be seen, there was a significant negative correlation between acid-soluble extractable metal and soil organic matter (SOM) and soil pH (Figure 3). There was also a significant negative correlation between the organic matter bound metal and SOM and pH, while the Fe and Mn oxide had an insignificant negative correlation for Cd and Cu. Regarding the Fe and Mn oxide for Pb, there was a significant negative correlation with SOM, while the Zn significantly positive correlation was observed with SOM and pH. In other words, based on the above results, in soil treated with biochar, acid-soluble extractable metal was decreased, and subsequently, the organic matter bound increased.

#### 3.4.6. Principal Component Analysis (PCA)

The PCA was used to determine the approximate correlation of the studied TCLP-extractable metals, extractable acid metals (F1), Fe and Mn oxides (F2), organic matter bound (F3) forms, soil pH, and soil organic matter (SOM) versus different biochar additives (Figure 4). The PCA shows that the total variance represents 89.7% for the TCPL-extractable metals, heavy metal forms, and their relation to pH and SOM. Regarding the results mentioned in the PCA confirm those obtained in our current study. Additionally, the organically bound realized while reducing the acid- extractable metal demonstrates the positive effect of the PB and BB treatments on reducing the studied metals’ risk.

## 4. Discussion

### 4.1. Biochar Type Effect

Biochar types affected soil properties (increased pH and OM and varied EC). The high soil pH and organic matter content (Table 2) might be attributed to biochar application. Similar results were obtained by [28,29]. The PB decreased the EC values, whereas the BB increased it as compared to the control. This may be attributed to salt adsorption, and its preservation resulted in a lower EC value of soil solution and/or movement of saline water downward, which reflected in a reduction EC value in the surface layer [2,28,30]. Our results clearly showed that all feedstock biochar reduced the risk of toxic metals by stabilizing them in insoluble forms. The PB was more effective in immobilizing these metals, especially at the highest level. This may be due to its high pH and EC values, which means a high elemental composition content as well as functional groups (Table 1, Figure 1 and Figure 2). These results are compatible with those obtained by [31,32,33]. They indicated that biochar’s elemental composition forms HM compounds with cations and anions via ion exchange, especially soft acids such as Pb^+2^ and Cu^+2^.

### 4.2. Biochar Rate Effect

The different rates of biochar application caused an increase of OM and alkalinity with few exceptions. The soil salinity (EC) was gradually increased by increasing BB rate, while it regularly decreased in PB. Similar results were observed by several authors [2,28,34]. The mobility reduction of the tested metals as a result of increasing biochar application was evident. This may be because the gradual addition of biochar resulted in increased pH values, which, in turn, was negatively reflected in the concentration of metals, especially in acidic soils. The increasing rate of biochar addition of bamboo and rice straw decreased HMs extractability, which was significantly highly (*p* < 0.01) correlated with soil pH [34]. Increased biochar additives might increase the formation of stable complexes via an increase in ions and functional groups. Raising soil pH in acidic soils due to increased biochar applications led to enhanced adsorption and reduced metal mobility [21,35]. The concentration of HMs extracted with calcium chloride (CaCl_2_) or DTPA of soil treated by biochar at the rate of 1 and 5% was significantly reduced at 5% compared to that at 1% [4,36].

### 4.3. Heavy Metals Mobility and Redistribution

The addition of biochar efficiently reduced the concentrations of soil TCLP and acid-soluble fractions of Pb, Cd, Zn, and Cu over untreated one (Table 3 and Table 4). This decrease may be due to the high pH and/or EC values of the applied biochar, as shown in Table 1, and their role in increasing the pH (especially in acidic soils) that reduced metal mobility [2,35]. The high surface adsorption capacity of biochar and/or formation of insoluble compounds could cause a reduction in the HMs solubility [2,37]. In the same context, [37] observed a significant reduction in the acid-soluble fraction of Pb, Cd, and Cu. In the current study, the BCR results indicated that the acid-soluble fraction for metals could be arranged in descending order of Cd > Zn >Pb > Cu. These results are in agreement with [38] who arranged the HMs according to acid-soluble fraction in descending order of Ni > Mn > Cd > Zn > Pb > Cr > Co > Cu. Soil amended biochar shows a large amount of Pb (60%, from the total amount of Pb) in Fe/Mn oxides bound fraction in comparison to other fractions. These results may be due to the affinity of Pb bound to Mn oxides. These results are compatible with those observed by [39], who extracted a significant amount of Pb (57.4%) from the Fe/Mn oxide bound fraction. Additionally, [40] found that Pb was the largest one associated with the oxides. However, the organically bound fraction increased induced biochar additions, while the soluble fraction decreased compared to the control. These decreases in the acid-soluble metals fraction were based on the increase of OM bound metal fraction. Moreover, the increase in the OM bound metal fraction was dependent on feedstock type and the level of addition. The biochar application caused an increase in the metal fraction bound to the organic matter, and the opposite trend was observed with soluble fraction [12,41]. Heavy metals could be adsorbed on biochar surfaces resulting in a reduction in their mobilization in biochar treated soils [42]. The data collected from correlation and PCA (Figure 3 and Figure 4) have further emphasized a strong relation between biochar additions and acid-soluble metals and SOM, pH, and metal bound to organic matter. Biochar additives resulted in a positive relationship with the organic matter, pH, and metals fraction associated with organic matter and a negative one with the acid-soluble fraction. This confirms the hypothesis that the increase in the organic matter caused an increase in the organic fraction (organic complexes) at the acid soluble fraction’s expense. Additionally, it caused an increase in pH, which leads us to confirm another hypothesis of metal precipitation as a normal result of increasing the pH.

### 4.4. Mechanisms of Heavy Metals Immobilization

Numerous studies, including our current study, have demonstrated the susceptibility of biochar to immobilize metals [2,12,41] but the mechanism is still unknown. One and/or more mechanisms are likely to be responsible for stabilizing metals in soils amended with biochar, among which are (1) natural adsorption on biochar surfaces [33], (2) chemical bonding with ions on its surface [32,33,43], (3) complexation with active functional groups [11,32], (4) precipitate via phosphates ions [11,31,44,45], and (5) precipitation by raising soil pH [13,46]. However, all these mechanisms are not compatible with all the metals studied due to the differences in the characteristics between these metals. While Cd and Zn tend to be physically adsorbed on the surface of biochar, Cu can bond chemically on the surface, and Pb can be complexed with a functional group and/or precipitate. In this study, the PB was more effective in HMs immobilization than BB. This is attributed mainly to the higher pH and EC values and/or functional groups of PB and its higher elemental composition content than BB one (Table 1, Figure 1 and Figure 2). It was found that the concentration of the TCLP-extractable metals was as follows: Cu (45%) >Zn (24%) > Pb (22%) > Cd (17%) due to the application of PB at the highest level. The decrease in portion extracted with acid and the increase in the organic matter bound fraction may explain the increase of organic matter bound metal fraction based on the reduction in acid extractable metals. PB had higher alkalinity, total dissolved salts, and O, Si, K, and Ca content than those of BB, which may explain the more significant reduction in HMs solubility. Organic amendments affect the mobility, and bioavailability of soil HMS depends on the specific metal, soil type, and the amendment characteristics such EC, CEC, and pH [30,47].

## 5. Conclusions

The risk of heavy metals increases in contaminated soil with the increase in the soluble form. Therefore, it became necessary to monitor the different changes of the metal forms and their relation to changes in soil properties after adding biochar. The results indicated that the redistribution of HMs mainly depends on the feedstock type (paulownia and bamboo), their application rate, soil pH, organic matter content, and metal type. The paulownia biochar was superior in terms of its effect on these metals’ solubility compared to bamboo biochar. The acid-soluble metals reduction was mostly at the expense of the organic matter and oxide pounds increases. High-significant negative correlations were noticed with both soil organic matter and pH from one side and the soluble fraction for all the studied metals from the other, while the opposite trend was observed with the organic matter bound fraction. The movement of HMs in soils and their transformations depends mainly on their geochemically related bond. This study proves that paulownia biochar is a good material for reducing heavy metals solubility to reduce hazardous environmental effects. Besides metal type, a reduction in HMs mobilization was dependent on raising the pH in acidic soils side by side with the increase in organic matter. Adding biochar is not a scientific curiosity but a useful technique to restrict metals’ mobility, consequently decreasing their uptake and improving soil properties.

## Figures and Tables

**Figure 1 biomolecules-11-00448-f001:**
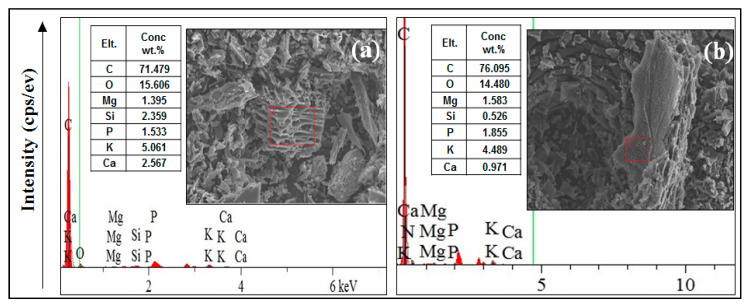
The energy dispersive X-ray spectra (EDS) collected from the scanning electron microscopy (SEM) regions of paulownia biochar (**a**), and bamboo biochar (**b**).

**Figure 2 biomolecules-11-00448-f002:**
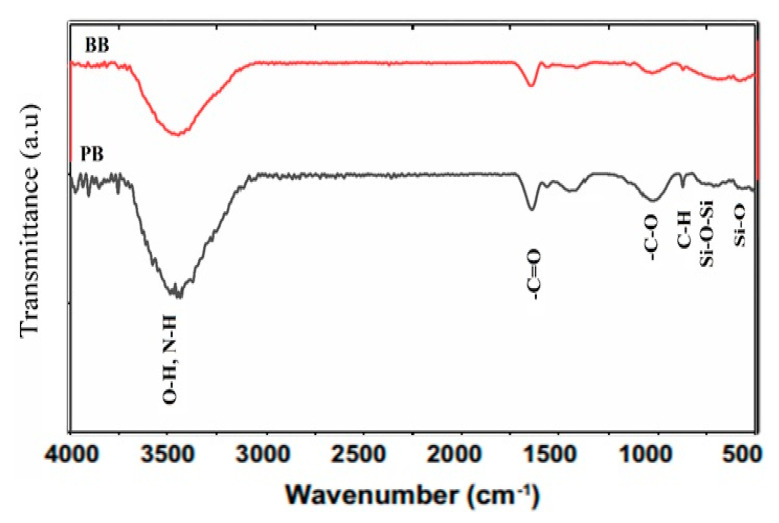
Fourier-transform infrared (FTIR) spectroscopy of tested biochar.

**Figure 3 biomolecules-11-00448-f003:**
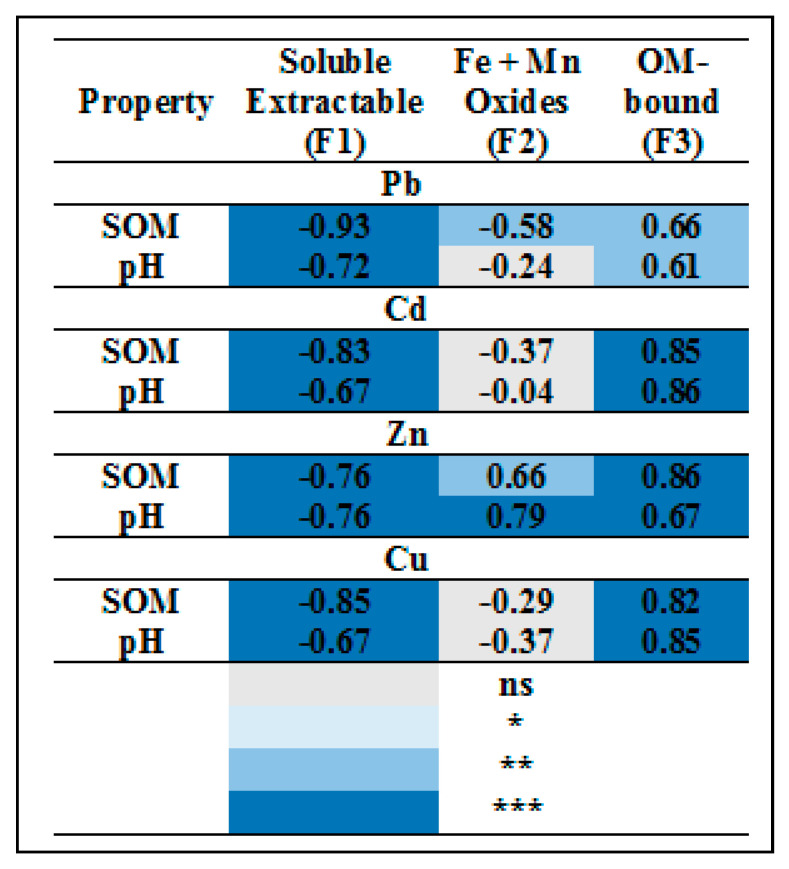
Values representing the correlation coefficient of the acid soluble extractable (F1), Fe and Mn oxides (F2), and organic matter bound (F3) in relation to soil organic matter (SOM) and soil pH. ns, Not significant, * significant, ** highly significant, *** very highly significant.

**Figure 4 biomolecules-11-00448-f004:**
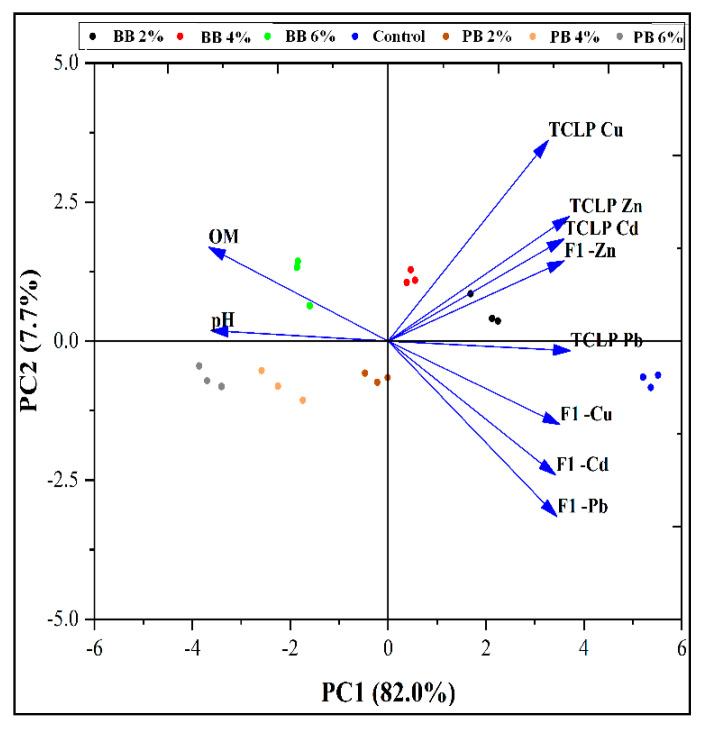
Principal component analysis of TCLP-extractable and acid soluble extractable metals in contaminated soil treated with paulownia biochar and bamboo biochar at the rate of (0-control, 2, 4, and 6%). PC: Principal Components.

**Table 1 biomolecules-11-00448-t001:** Some characteristics of the tested soil and the utilized biochar.

Property	Unit	Soil	Biochar	
			Paulownia	Bamboo
Clay	(g kg^−1^)	260	-	-
Silt	(g kg^−1^)	320	-	-
Sand	(g kg^−1^)	420	-	-
Texture		loam	-	-
Organic matter	(g kg^−1^)	24.40	840	732
EC (1:5)	(dS m^‒1^)	0.90	1.80	0.75
pH (1:2.5)		5.50	10.50	10.00
Total Cd	(mg kg^−1^)	2.50	ND *	ND *
Total Pb	(mg kg^−1^)	970	4.00	ND *
Total Zn	(mg kg^−1^)	1010	40.00	30.50
Total Cu	(mg kg^−1^)	36	15.50	3.88

* Not detected, each value represents the mean of three replicates.

**Table 2 biomolecules-11-00448-t002:** Effect of biochar application on soil organic matter (%), soil pH (1:2.5), and electrical conductivity (EC, dSm^−1^).

Biochar Type	Biochar Rate	OM	pH	EC
CK	0	2.40 ± 0.05 d	5.54 ± 0.10 d	0.64 ± 0.10 b
PB	2	3.58 ± 0.04 c	5.75 ± 0.07 c	0.34 ± 0.09 c
	4	4.31 ± 0.30 b	6.21 ± 0.11 a	0.38 ± 0.11 c
	6	5.47 ± 0.04 a	6.048 ± 0.7 ab	0.39 ± 0.08 c
BB	2	3.51 ± 0.16 c	5.58 ± 0.10 d	0.65 ± 0.10 b
	4	4.28 ± 0.23 b	5.92 ± 0.08 b	0.68 ± 0.10 b
	6	5.21 ± 0.16 a	6.05 ± 0.10 b	0.76 ± 0.08 a

All values are the means of triplicate analysis. Within each column, values followed by the same letter are not significantly different at *p* < 0.05 level. CK, control (no biochar applied); PB, Paulownia biochar; BB, Bamboo biochar.

**Table 3 biomolecules-11-00448-t003:** Effects of biochar addition on concentration (mg kg^−1^) of the TCLP-extractable metals in the contaminated soil.

Biochar Type	Biochar Rate	Pb	Cd	Zn	Cu
CK	0	49.95 ± 0.63 a	1.45 ± 0.20 a	132.03 ± 1.77 a	0.850 ± 0.08 a
PB	2	42.06 ± 0.58 c	1.26 ± 0.08 c	113.51 ± 1.88 d	0.698 ± 0.05 c
	4	40.01 ± 0.76 d	1.23 ± 0.09 d	107.63 ± 1.67 f	0.546 ± 0.08 d
	6	38.90 ± 0.35 d	1.21 ± 0.10 e	99.94 ± 1.26 g	0.470 ± 0.07 e
BB	2	48.34 ± 0.44 b	1.40 ± 0.08 b	128.10 ± 1.23 b	0.774 ± 0.08 b
	4	42.51 ± 0.44 c	1.39 ± 0.11 b	123.98 ± 1.78 c	0.774 ± 0.03 b
	6	39.73 ± 0.51 d	1.24 ± 0.09 d	112.00 ± 1.43 e	0.749 ± 0.07 b

All values are the means of triplicate analysis. Within each column, values followed by the same letter are not significantly different at *p* < 0.05 level. CK, control (no biochar applied); PB, Paulownia biochar; BB, Bamboo biochar.

**Table 4 biomolecules-11-00448-t004:** Concentration of the acid extractable fraction of soil heavy metals (mg kg^−1^) determined with the BCR sequential extraction method.

Biochar Type	Biochar Rate	Pb	Cd	Zn	Cu
CK	0	137.46 ± 0.05 a	1.85 ± 0.04 a	168.01 ± 1.00 a	2.05 ± 0.08 a
PB	2	126.45 ± 0.034 b	1.71 ± 0.04 b	145.15 ± 3.00 c	1.65 ± 0.7 b
	4	125.46 ± 0.05 c	1.64 ± 0.03 bc	144.92 ± 4.00 c	1.53 ± 0.5 b
	6	123.48 ± 1.0 d	1.60 ± 0.02 c	141.75 ± 1.00 c	1.27 ± 0.05 c
BB	2	129.13 ± 0.58 e	1.68 ± 0.05 b	155.84 ± 1.00 b	1.66 ± 0.08 b
	4	124.50 ± 0.07 f	1.66 ± 0.06 bc	153.59 ± 1.00 b	1.66 ± 0.07 b
	6	120.14 ± 0.55 g	1.64 ± 0.04 bc	151.79 ± 2.00 b	1.41 ± 0.06 bc

All values are the means of triplicate analysis. Within each column, values followed by the same letter are not significantly different at *p* < 0.05 level. CK, control (no biochar applied); PB, Paulownia biochar; BB, Bamboo biochar.

**Table 5 biomolecules-11-00448-t005:** Concentration of the soil heavy metal fraction bound to Fe/Mn oxides (mg kg^−1^) determined with the BCR sequential extraction method.

Biochar Type	Biochar Rate	Pb	Cd	Zn	Cu
CK	0	609.33 ± 16 a	0.491 ± 0.05 ab	85.16 ± 2.00 c	6.71 ± 0.13 bc
PB	2	549.45 ± 6 d	0.487 ± 0.06 ab	89.04 ± 1.00 b	6.45 ± 0.35 cd
	4	592.63 ± 5 ab	0.498 ± 0.03 a	97.89 ± 2.00 a	6.25 ± 0.05 d
	6	557.78 ± 9 cd	0.477 ± 0.02 ab	97.76 ± 3.00 a	5.85 ± 0.07 e
BB	2	576.00 ± 3 bc	0.481 ± 0.04 ab	90.77 ± 2.00 b	7.14 ± 0.22 a
	4	572.67 ± 1 bc	0.488 ± 0.05 ab	94.60 ± 1.00 a	7.04 ± 0.05 ab
	6	526.18 ± 15 e	0.463 ± 0.05 b	90.25 ± 1.00 b	7.05 ± 0.08 ab

All values are the means of triplicate analysis. Within each column, values followed by the same letter are not significantly different at *p* < 0.05 level. CK, control (no biochar applied); PB, Paulownia biochar; BB, Bamboo biochar.

**Table 6 biomolecules-11-00448-t006:** Concentration of the soil heavy metal fraction bound to organic matter (mg kg^−1^) determined with the BCR sequential extraction method.

Biochar Type	Biochar Rate	Pb	Cd	Zn	Cu
CK	0	61.70 ± 2 c	0.104 ± 0.02 d	140.82 ± 1.00 f	7.34 ± 0.08 d
PB	2	91.33 ± 2 c	0.147 ± 0.05 bc	165.67 ± 2.00 e	9.05 ± 0.22 c
	4	98.62 ± 2 a	0.164 ± 0.06 a	185.11 ± 3.00 c	10.45 ± 0.38 b
	6	99.96 ± 4 a	0.168 ± 0.07 a	189.35 ± 2.00 b	11.80 ± 0.58 a
BB	2	98.96 ± 2 a	0.144 ± 0.08 c	172.41 ± 1.00 d	8.49 ± 0.31 c
	4	99.49 ± 3 a	0.147 ± 0.08 bc	189.90 ± 1.00 b	8.79 ± 0.22 c
	6	99.63 ± 3 a	0.155 ± 0.05 b	200.01 ± 1.00 a	8.93 ± 0.08 c

All values are the means of triplicate analysis. Within each column, values followed by the same letter are not significantly different at *p* < 0.05 level. CK, control (no biochar applied); PB, Paulownia biochar; BB, Bamboo biochar.

**Table 7 biomolecules-11-00448-t007:** Concentration of the residual fraction of soil heavy metals (mg kg^−1^) determined with the BCR sequential extraction method.

Biochar Type	Biochar Rate	Pb	Cd	Zn	Cu
CK	0	195.95 ± 2.00 a	0.127 ± 0.05 b	658.00 ± 15 a	16.89 ± 0.10 a
PB	2	186.60 ± 2.00 bcd	0.146 ± 0.06 a	645.60 ± 7 a	15.86 ± 0.11 a
	4	193.53 ± 3.00 ab	0.146 ± 0.08 a	667.15 ± 2 a	15.79 ± 0.11 a
	6	180.81 ± 5.00 d	0.146 ± 0.07 a	649.75 ± 2 a	15.25 ± 1.57 a
BB	2	192.06 ± 4.00 abc	0.145 ± 0.04 a	629.78 ± 3 a	15.92 ± 0.33 a
	4	185.10 ± 4.00 cd	0.141 ± 0.06 a	667.77 ± 10 a	15.31 ± 0.81 a
	6	173.99 ± 2.00 e	0.140 ± 0.03 a	664.73 ± 11 a	15.63 ± 0.88 a

All values are the means of triplicate analysis. Within each column, values followed by the same letter are not significantly different at *p* < 0.05 level. CK, control (no biochar applied); PB, Paulownia biochar; BB, Bamboo biochar.

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
