# Peer review of "Fractionation of Heavy Metals in Multi-Contaminated Soil Treated with Biochar Using the Sequential Extraction Procedure"

_biomolecules, 2021, doi:10.3390/biom11030448_

Round 1

Reviewer 1 Report

Presented manuscript is another study of biochar effects on heavy metal immobilization in soil showing very similar results to other already published papers on the topic and in my opinion novelty of research is limited. All the elements of the manuscript are designed in a proper way and clearly presented. Some more detailed comments are given in the pdf file.  

Author Response

Reviewer #1:

Authors’ response: Thank you so much for your valuable remarks, which have allowed publishing in the journal. We also inform that we have been rechecked the whole manuscript and have edited where necessary. Please check all edits as track change mode.

  • Please provide more detailed information about the area where the soil samples were collected. Zn and Pb contents are extremely high, hopefully this is not a typical situation for Shaoquan city.

Authors response: we provide more detailed

Surface soil samples (0-20 cm) were collected from Shaoquan city, Guangdong province, north China, which is a representative area for mining activities. Also, it receives liquid discharged wastes for more than 50 years, that causes a high heavy metals level, especially lead and zinc. The collected samples air-dried, crushed, and sieved through 2mm sieve, and then kept for soil analysis

  • Line 153-154, The illustrated data in Table 3 shows that the biochar additions caused a decrease in HMs concentration than the CK treatment.

Authors response: compared with CK treatment"

  • Figure 2, This figure is too big and the quality is low, please provide a high-resolution figure

Authors response: we improved it

  • Figure 3, This figure is too big and the quality is low, please provide a high-resolution figure

Authors response: we improved it

  • Can authors prove this hypothesis? Any data about Fe and Mn content in biochars can be provided? Authors claimed before that Pb was mainly bound in organic matter fraction when biochar was applied to soil (lines 209-210)

Authors response Line 209-210: Based on the data in the table, this result is not an allegation but a fact, as the addition of biochar increased the part associated with the organic matter by about 60% compared to the control.

As for line 395-396, this sentence means that 60% of the native amount of lead present in the soil was associated with iron and manganese oxides in comparisom to other fractions.

There is no conflict between the two sentences

  • Please rewrite this sentence "heavy metals can be made stable" is not correct in English

Authors response; We rewrote it,

Heavy metals could be adsorbed on biochar surfaces resulting in a reduction in their mobilization in biochar treated soils

  • Please explain why? Authors mentioned all known mechanisms of heavy metal immobilization by biochar.

Authors response; However, these all mechanisms are not compatible with all studied metals due to the differences in the characteristics between these metals.

  • References 18, 19 and 20 are different style

Authors response; we made it in the same format.

Reviewer 2 Report

The manuscript is interesting, well ordered and structured. However, some changes/improvements are necessary before recommending its publication.

According to the paper the biochars studied increased the soil pH and SOM content and thus reduced the concentration of metals in the more mobile phases. But, this result is not new. Authors should highlight/explain the novelty of this manuscript in the introduction (and mention in the abstract and conclusión sections).

In relation to the mechanisms, in section 4.5 it is said "This is attributed mainly to the higher pH and EC values and/or functional groups of PB and its higher elemental composition content than BB one (Table 1 and Fig. 1)" but the characterizatin of the chars is really poor. You mean functional groups... which functional groups? Have you studied them? (FT-IR or 13C NMR could give some information about that. Please see https://doi.org/10.3390/su12156025, and references threin, which could be also informative to complete the introduction section.

Authors rely on the elemental composition of the biochars, but the analysis is very poor. They have NOT determined the carbon content (I mean it is presented as OM), they have not determined the ash content nor the H content, which is fundamental to see the H/C ratio and therefore the aromaticity of the biochars.

They only study the surface by studying a small area by EDS, if the biochar is very heterogeneous this is not representative. In fact it is observed in the SEM that biochars are very heterogeneous.

The analysis of Porosity and SSA measurements should be included. 

Other small corrections needed.

-Introduce the achronyms before its use.

-Line 27: PB instead of Pb.

-Check gaps between no. and % (e.g. line 27 and line 32) or between no. and unit, ºC (line 91).

- Section 2.1

Add grain size of biochar. Add pyrolysis time, reactor type,

Include soil classification,

Provide names of equipment and brands used EDS, pH...

-Section 2.3

They explain in great detail the sequential extraction, which is already explained in the BCR reference. Indicate the complete citation (line 101).

-In all tables check the decimal places and provide SD (or standard error of the triplicate analysis). Provide the units of all the measurements (headings of each table).

-Figure 2. Quality and footnote "ns...*...***...".

-Section 4.1

Please rewrite the last part. (High content of the elemental composition?-line 261-262)

Revise spaces >Pb>.... (line 290)

Conclusions section

The conclusions sectin should be re-written. The present version is too general, they do not seem to be conclusions of this work. Please, focus on making conclusions from the results of this study and be more precise.

(For instance...: Pawlonia biochar is a good material…. (provide further details of how you got such a conclusión)

Author Response

Reviewer #2:

Authors’ response: Thank you so much for your appreciated remarks, which have permitted to consider the findings to publish in the journal. We are to inform that we have been able to address all of your valuable suggestions. Besides these we have also rechecked the whole manuscript and have edited where necessary. Please check all edits as track change mode.

  • According to the paper the biochars studied increased the soil pH and SOM content and thus reduced the concentration of metals in the more mobile phases. But, this result is not new. Authors should highlight/explain the novelty of this manuscript in the introduction (and mention in the abstract and conclusión sections).

Authors response: We did

  • In relation to the mechanisms, in section 4.5 it is said "This is attributed mainly to the higher pH and EC values and/or functional groups of PB and its higher elemental composition content than BB one (Table 1 and Fig. 1)" but the characterizatin of the chars is really poor. You mean functional groups... which functional groups? Have you studied them? (FT-IR or 13C NMR could give some information about that. Please see https://doi.org/10.3390/su12156025, and references threin, which could be also informative to complete the introduction section.

Authors response: In we add FTIR analysis

  • Line 27: PB instead of Pb.

Authors response: We correct it

  • -Check gaps between no. and % (e.g. line 27 and line 32) or between no. and unit, ºC (line 91).

Authors response: We checked it

  • In section 2.1, Add grain size of biochar. Add pyrolysis time, reactor type, Include soil classification, Provide names of equipment and brands used EDS, pH...

Authors response: We add it

  • Section 2.3, They explain in great detail the sequential extraction, which is already explained in the BCR reference. Indicate the complete citation (line 101).

Authors response: The complete citation found was provided in the sequential extraction procedure section Line 114   "A sequential extraction procedure was applied according to EUBCR to determine heavy metals redistribution (Pb, Cd, Zn and Cu) as described by [26]".

  • In all tables check the decimal places and provide SD (or standard error of the triplicate analysis). Provide the units of all the measurements (headings of each table).

Authors response: We checked it

  • Figure 2. Quality and footnote "ns...*...***...".

Authors response: We improved it.

  • Please rewrite the last part. (High content of the elemental composition?-line 261-262)

Authors response: We rewrote it.

  • Revise spaces > Pb >.... (line 290)

Authors response: We revised it.

This manuscript is a resubmission of an earlier submission. The following is a list of the peer review reports and author responses from that submission.

Round 1

Reviewer 1 Report

The authors report a study in which biochar was added to a soil, incubated for two months at 60-70% field capacity, sequentially extracted metal fractions and attributed the outcome to various mineral phases.

First, no international classification is provided for the soils.  

Second, biochar produced by high-temperature pyrolysis contains fractions that react with chromate though they may not be equivalent to soil organic matter. 

Third, the attribution of the results of sequential extractions to specific mineral phases is discredited and such attributions are poorly associated with bioavailability.

Lastly, any sorption effects are confounded with those on pH.

Reviewer 2 Report

  • Title – current version is very condensed. Authors tried to put all information into one sentence and as a result title is not catchy. It suggest different methodological approach rather than analyzing influence of biochar on bioavailability of heavy metals.
  • Abstract – What is the significance of “62.02%” – it is just 62%. I would suggest to round those numbers.
  • Abstract – authors provided a rather extensive but also descriptive sentences. I would suggest to make it a bit shorter. Additionally please add 1-2 sentences to provide a bigger view like summary – a message that authors would like to share with all readers. What is the influence of biochar on bioavailability ? Does it have any significance ? Can we make a good use of it ? What are the future considerations ? Abstract starts with following sentence: “The toxicity of heavy metals (HMs) represents a global problem depending on their geochemical forms in the soil environment.” Thus please finish this section with a sentence that would correspond to the sentence that opened the section.
  • Introduction – lines 36-41 – I am not sure about such a fast introduction of specific city. The flow of the text is lost here. Generally the whole section looks like a patch-work. Different issues are glued together but the flow is lost. Please introduce the problem, present the field which needs to be researched in more details. Presents what is known and what needs to be further investigated. Present a scientific hypothesis and briefly show how you are going to verify your hypothesis. It is not a difficult task but order of sentences need to be changed as well as some additional information need to be added. Since there are several reports on biochars etc. – it should be a relatively easy task.
  • Figure 2 is blurry
  • Figure 3 is too big, subtitles are not unified with the text font. The difference between manuscript published in top ranked journals and any other journal is usually visible as the quality of graphs, plots and tables. It is easy to work with it a bit and make it really nice. Time will be paid off in the higher number of citation.
  • Check the whole manuscript and unify the way the units are given – sometimes spacebar is used between degrees symbol sometimes not – just make it uniform.
  • I would suggest to have raw data in the tables but use rounded numbers in the text – it does not really matter if we have “14.03 to 16.63%” or 14-17%. Trends are more important, besides for such a precision more samples are needed together with decent statistical treatment of the data.
  • Discussion – I would expect to find a general summary of the research – not only interpretation of the data but final statements. How can we apply biochars to make our environment a better place ? Is it plausible ? At what cost ? Is it a scientific curiosity or a potentially useful technology ?